# MACHINE LEARNING FORCE FIELDS WITH DATA COST AWARE TRAINING

## ABSTRACT

Machine learning force fields (MLFF) have been proposed to accelerate molecular dynamics (MD) simulation, which finds widespread applications in chemistry and biomedical research. Even for the most data-efficient MLFF models, reaching chemical accuracy can require hundreds of frames of force and energy labels generated by expensive quantum mechanical algorithms, which may scale as $O(n^3)$ to $O(n^7)$, with $n$ being the number of basis functions used and typically proportional to the number of atoms. To address this issue, we propose a multi-stage computational framework – ASTEROID, which enjoys low training data generation cost without significantly sacrificing MLFFs' accuracy. Specifically, ASTEROID leverages a combination of both large cheap inaccurate data and small expensive accurate data. The motivation behind ASTEROID is that inaccurate data, though incurring large bias, can help capture the sophisticated structures of the underlying force field. Therefore, we first train a MLFF model on a large amount of inaccurate training data, employing a bias-aware loss function to prevent the model from overfitting the potential bias of the inaccurate training data. We then fine-tune the obtained model using a small amount of accurate training data, which preserves the knowledge learned from the inaccurate training data while significantly improving the model's accuracy. Moreover, we propose a variant of ASTEROID based on score matching for the setting where the inaccurate training data are unlabelled. Extensive experiments on MD simulation datasets show that ASTEROID can significantly reduce data generation costs while improving the accuracy of MLFFs.

## 1 INTRODUCTION

Molecular dynamics (MD) simulation is a key technology driving scientific discovery in fields such as chemistry, biophysics, and materials science (Alder & Wainwright, 1960; McCammon et al., 1977). By simulating the dynamics of molecules, important macro statistics such as the folding probability of a protein (Tuckerman, 2010) or the density of new materials (Varshney et al., 2008) can be estimated. These macro statistics are an essential part of many important applications such as structure-driven drug design (Hospital et al., 2015) and battery development (Leung & Budzien, 2010). Most MD simulation techniques share a common iterative structure: MD simulations calculate the forces on each atom in the molecule, and use these forces to move the molecule forward to the next state.

The fundamental challenge of MD simulation is how to efficiently calculate the forces at each iteration. An exact calculation requires solving the Schrödinger equation, which is not feasible for many-body systems (Berezin & Shubin, 2012). Instead approximation methods such as the Lennard-Jones potential (Johnson et al., 1993), Density Functional Theory (DFT, Kohn (2019)), or Coupled Cluster Single-Double-Triple (CCSD(T), Scuseria et al. (1988)) are used. CCSD(T) is seen as the gold-standard for force calculation, but is computationally expensive. In particular, CCSD(T) has complexity $\mathcal{O}(n^7)$ with respect to the number of basis function used along with a huge storage requirement (Chen et al., 2020). To accelerate MD simulation while maintaining high accuracy, machine learning based force fields have been proposed. These machine learning models take a molecular configuration as input and then predict the forces on each atom in the molecule. These models have been successful, producing force fields with moderate accuracy while drastically reducing computation time (Chmiela et al., 2017).

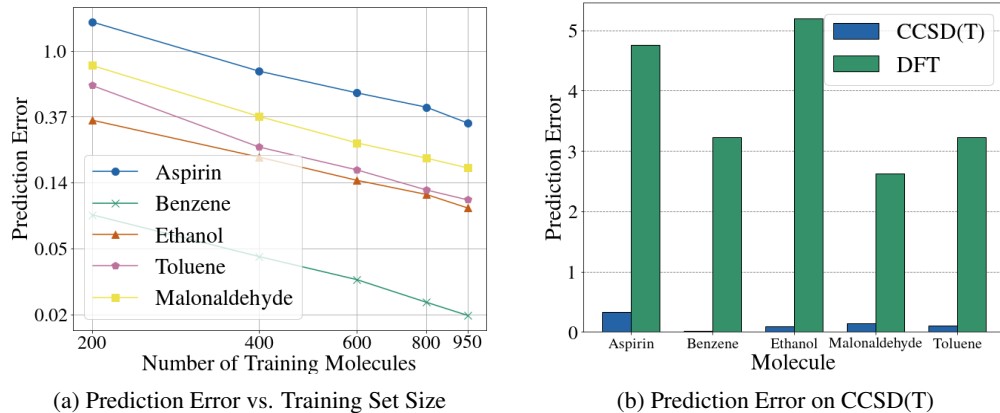

(a) Prediction Error vs. Training Set Size          (b) Prediction Error on CCSD(T)

Figure 1: (a) Log-log plot of the number of training point versus the prediction error for deep force fields (b) Prediction error on CCSD labelled molecules for force fields trained on large amounts of DFT reference forces (100,000 configurations) and moderate amounts of CCSD reference forces (1000 configurations). In both cases the model architecture used is GemNet (Gasteiger et al., 2021).

Built upon the success of machine learning force fields, deep learning techniques for force fields have been developed, resulting in highly accurate force fields parameterized by large neural networks (Gasteiger et al., 2021; Batzner et al., 2022). Despite this empirical success, a key drawback is rarely discussed in existing literature: *in order to train state-of-the-art machine learning force field models, a large amount of costly training data must be generated*. For example, to train a model at the CCSD(T) level of accuracy, at least a thousand CCSD(T) calculations must be done to construct the training set. This is computationally expensive due to the method's $\mathcal{O}(n^7)$ cost.

A natural solution to this problem is to train on less data points. However, if the number of training points is decreased, the accuracy of the learned force fields quickly deteriorates. In our experiments, we empirically find that the prediction error and the number of training points roughly follow a power law relationship, with prediction error $\sim \frac{1}{\text{Number of Training Points}}$. This can be seen in Figure 1a, where prediction error and train set size are observed to have a linear relationship with a slope of $-1$ when plotted on a log scale.

Another option is to train the force field model on less accurate but computationally cheap reference forces calculated using DFT (Kohn, 2019) or empirical force field methods (Johnson et al., 1993). However, these algorithms introduce undesirable bias into the force labels, meaning that the trained models will have poor performance. This phenomenon can be seen in Figure 1b, where models trained on large quantities of DFT reference forces are shown to perform poorly relative to force fields trained on moderate quantities of CCSD(T) reference forces. Therefore current methodologies are not sufficient for training force fields models in low resource settings, as training on either small amounts of accurate data (i.e. from CCSD(T)) or large amounts of inaccurate data (i.e. from DFT or empirical force fields) will result in inaccurate force fields.

To address this issue, we propose to use both large amounts of inaccurate force field data (i.e. DFT) and small amounts of accurate data (i.e. CCSD(T)) to significantly reduce the cost of the data needed to achieve highly accurate force fields. Our motivation is that computationally cheap inaccurate data, though incurring large bias, can help capture the sophisticated structures of the underlying force field. Moreover, if treated properly, we can further reduce the bias of the obtained model by taking advantage of the accurate data.

More specifically, we propose a multi-stage computational framework – dat**A** cos**ST** awar**E** t**R**aining of f**O**rce f**I**el**D**s (ASTEROID). In the first stage, small amounts of accurate data are used to identify the bias of force labels in a large but inaccurate dataset. In the second stage, the model is trained on the large inaccurate dataset with a bias aware loss function. Specifically, the loss function generates smaller weights for data points with larger bias, suppressing the effect of label noise on training. This inaccurately trained model serves as a warm start for the third stage, where the force field model is fine-tuned on the small and accurate dataset. Together, these stages allow the model to learn

from many molecular configurations while incorporating highly accurate force data, significantly outperforming conventional methods trained with similar data generation budgets.

Beyond using cheap labelled data to boost model performance, we also extend our method to the case where a large amount of unlabelled molecular configurations are cheaply available (Smith et al., 2017; Köhler et al., 2022). Without labels, we cannot adopt the supervised learning approach. Instead, we draw a connection to score matching, which learns the gradient of the log density function with respect to each data point (called the score) (Hyvärinen & Dayan, 2005). In the context of molecular dynamics, we notice that if the log density function is proportional to the energy of each molecule, then the score function with respect to a molecule's position is equal to the force on the molecule. Based on this insight, we show that the supervised force matching problem can be tackled in an unsupervised manner. This unsupervised approach can then be incorporated into the ASTEROID framework, improving performance when limited data is available.

We demonstrate the effectiveness of our framework with extensive experiments on different force field data sets. We use two popular model architectures, GemNet (Gasteiger et al., 2021) and EGNN (Satorras et al., 2021), and verify the performance of our method in a variety of settings. In particular, we show that our multi-stage training framework can lead to significant gains when either DFT reference forces or empirical force field forces are viewed as inaccurate data and CCSD(T) configurations are used as accurate data. In addition, we show that we can learn accurate forces via the connection to score matching, and that using this objective in the second stage of training can improve performance on both DFT and CCSD(T) datasets. Finally, our framework is backed by a variety of ablation studies.

The rest of this paper is organized as follows: Section 2 presents necessary background on machine learning force fields and training data generation, Section 3 details the ASTEROID framework, Section 4 extends ASTEROID to settings where unlabelled configurations are available, Section 5 presents our experimental results, and Section 6 compares our method to several related works (Ramakrishnan et al., 2015; Ruth et al., 2022; Nandi et al., 2021; Smith et al., 2019; Deringer et al., 2020) and briefly concludes the paper.

## 2 BACKGROUND

◇ **Machine Learning Force Fields.** Recent years have seen a surge of interest in machine learning force fields. Much of this work has focused on developing large machine learning architectures that have physically correct equivariances, resulting in large graph neural networks that can generate highly accurate force and energy predictions (Gasteiger et al., 2021; Satorras et al., 2021; Batzner et al., 2022). Two popular architectures are EGNN and GemNet. Both models have graph neural network architectures that are translation invariant, rotationally equivariant, and permutation equivariant. EGNN is a smaller model and is often used when limited resources are available. The GemNet architecture is significantly larger and more refined than the EGNN architecture, modeling various types of inter-atom interactions. GemNet is therefore more powerful and can achieve state-of-the-art performance, but requires more resources.

It has been noted that in order to be reliable enough for MD simulations, a molecular force field must attain an accuracy of $1 \, \text{kcal} \, \text{mol}^{-1} \, \text{Å}^{-1}$ (Chmiela et al., 2018). Critically, the accuracy of deep force fields such as GemNet and EGNN is highly dependent on the size and quality of the training dataset. With limited training data, modern MLFFs cannot achieve the required accuracy for usefulness, preventing their application in settings where data is expensive to generate (e.g. large molecules). The amount of resources needed to train is therefore a key bottleneck preventing widespread use of modern MLFFs.

◇ **Data Generation Cost.** The training data for MLFFs can be generated by a variety of force calculation methods. These methods exhibit an accuracy cost tradeoff: accurate reference forces from methods such as CCSD(T) require high computational costs to generate reference forces, while inaccurate reference forces from methods such as DFT and empirical force fields can be generated extremely quickly. Concretely, CCSD(T) is highly accurate but has $\mathcal{O}(n^7)$ complexity, DFT is less accurate with complexity $\mathcal{O}(n^3)$, and empirical force fields are inaccurate but have complexity $\mathcal{O}(n)$ (Lin et al., 2019; Ratcliff et al., 2017). CCSD(T) is typically viewed as the gold standard for calculating reference forces, but its computational costs often make it impractical for MD simulation (it has been estimated that "a nanosecond-long MD trajectory for a single ethanol molecule exe-

cuted with the CCSD(T) method would take roughly a million CPU years on modern hardware")
(Chmiela et al., 2018). Due to this large expense, molecular configurations are typically generated
first with MD simulations driven by DFT or empirical force fields. These simulations generate a
large number of molecular configurations, and then CCSD(T) reference forces are computed for
a small portion of these configurations. Due to this generation process, typically, a large amount
of inaccurately labelled molecular configurations are available along with the accurate CCSD(T)
labelled data. Therefore large amounts of inaccurate force data can be obtained and used to train
MLFF models, while incurring almost no extra data generation costs compared to CCSD(T).

## 3    ASTEROID

To reduce the amount of resources needed to train machine learning force fields, we propose a
multi-stage training framework, ASTEROID, to learn from a combination of both cheaply available
inaccurate data and more expensive accurate data.

**Preliminaries.** For a molecule with $k$ atoms, we denote a configuration (the positions of its atoms
in 3D) of this molecule as $x \in \mathbb{R}^{3k}$, its respective energy as $E(x) \in \mathbb{R}$, and its force as $F(x) \in \mathbb{R}^{3k}$. We denote the accurately labelled data as $\mathcal{D}_A = \{(x_1^a, e_1^a, f_1^a), ..., (x_N^a, e_N^a, f_N^a)\}$ and the
inaccurately labelled data as $\mathcal{D}_I = \{(x_1^n, e_1^n, f_1^n), ..., (x_M^n, e_M^n, f_M^n)\}$, where $(x_i^a, e_i^a, f_i^a)$ represents
the position, potential energy, and force of the $i$th accurately labelled molecule, using shorthand
notations $e_i^a = E(x_i^a)$ and $f_i^a = F(x_i^a)$ (similarly $(x_j^n, e_j^n, f_j^n)$ for the $j$th inaccurately labeled
data). Conventional methods train a force field model $E(\cdot; \theta)$ with parameters $\theta$ on the accurate data
by minimizing the loss

$$\min_\theta \mathcal{L}(\mathcal{D}_A, \theta) = (1 - \rho) \sum_{i=1}^N \ell_f(f_i^a, \nabla_x E(x_i^a; \theta)) + \rho \sum_{i=1}^N \ell_e(e_i^a, E(x_i^a; \theta)), \quad (1)$$

where $\ell_f$ is the loss function for the force prediction, and $\ell_e$ is the loss function for the energy
prediction. Here the force is denoted by $\nabla_x E(x; \theta)$, i.e., the gradient of the energy $E(x; \theta)$ w.r.t. to
the input $x$. We can use $\rho$ to balance the losses of the energy prediction and the force prediction. In
practice, most of the emphasis is placed on the force prediction, e.g. $\rho = 0.001$.

### 3.1    BIAS IDENTIFICATION

The approximation algorithms used to generate cheap data $\mathcal{D}_I$ introduce a large amount of bias
into some force labels $f^n$, which may significantly hurt training. Motivated by this phenomenon,
we aim to identify the most biased force labels so that we can avoid overfitting to the bias during
training. In the first stage of our training framework, we use small amounts of accurately labelled
data $\mathcal{D}_A$ to identify the levels of bias in the inaccurate dataset $\mathcal{D}_I$. Specifically, we train a force
field model by minimizing $\mathcal{L}(\mathcal{D}_A, \theta)$ (Eq. 1), the loss over the accurate data, to get parameters
$\theta_0$. Although the resulting model $E(\cdot; \theta_0)$ will not necessarily have good prediction performance
because of the limited amount of training data, it can still help estimate the bias of the inaccurate
data. For every configuration $x_j^n$ in the inaccurate dataset $\mathcal{D}_I$, we suspect it to have a large bias if
there is a large discrepancy between its force label $f_j^n$ and the force label predicted by the accurately
trained model $\nabla_x E(x_j^n; \theta_0)$. We can therefore use this discrepancy as a surrogate for bias, i.e.
$B(x_j^n) = \|\nabla_x E(x_j^n; \theta_0) - f_j^n\|_1$.

### 3.2    BIAS-AWARE TRAINING WITH INACCURATE DATA

In the second stage of our framework, we train a force field model $E(\cdot; \theta_{\text{init}})$ from scratch on large
amounts of *inaccurately labelled data* $\mathcal{D}_I$. To avoid over-fitting to the biased force labels, we use
a bias aware loss function that weighs the inaccurate data according to their bias. In particular, we
use the weights $w_j = \exp(-B(x_j^n)/\gamma)$ for configuration $x_j^n$, where $\gamma$ is a hyperparameter to be
tuned. In this way, low-bias points are given higher importance and high bias points are treated
more carefully. We then minimize the bias aware loss function

$$\min_\theta \mathcal{L}_w(\mathcal{D}_I, \theta) = (1 - \rho) \sum_{i=1}^M w_i \cdot \ell_f(f_i^n, \nabla_x E(x_i^n; \theta)) + \rho \sum_{i=1}^M w_i \cdot \ell_e(e_i^n, E(x_i^n; \theta)), \quad (2)$$

to get parameters $\theta_{\text{init}}$, resulting in the initial estimate of the force field model $E(\cdot; \theta_{\text{init}})$.

### 3.3 FINE-TUNING OVER ACCURATE DATA

The model $E(\cdot; \theta_{\text{init}})$ contains information useful to the force prediction problem, but may still contain bias because it is trained on inaccurately labelled data $\mathcal{D}_I$. Therefore, we further refine it using accurately labelled data $\mathcal{D}_A$. Specifically, we use $E(\cdot; \theta_{\text{init}})$ as an initialization for our final stage, in which we fine-tune the model over the accurate data by minimizing $\mathcal{L}(\mathcal{D}_A, \theta_{\text{final}})$ (Eq. 1).

## 4 ASTEROID FOR UNLABELLED DATA

In several settings molecular configurations are generated without force labels, either because they are not generated via MD simulation (e.g. normal mode sampling, Smith et al. (2017)) or because the forces are not stored during the simulation (Köhler et al., 2022). Although these unlabelled configurations may be cheaply available, they are not generated for the purpose of learning force fields and have not been used in existing literature. Here, we show that the unlabelled configurations can be used to obtain an initial estimate of the force field, which can then be further fine-tuned on accurate data. More specifically, we consider a molecular system where the number of particles, volume, and temperature are constant (NVT ensemble). Let $x$ refer to the molecule's configuration and $E(x)$ refer to the corresponding potential energy. It is known that $x$ follows a Boltzmann distribution, i.e.

$$p(x) = \frac{1}{Z}\exp\left(-\frac{1}{k_\beta T}E(x)\right),$$

where $Z$ is a normalizing constant, $T$ is the temperature, and $k_\beta$ is the Boltzmann constant. In practice, configurations generated using normal mode sampling (Unke et al., 2021) or via a sufficiently long NVT MD simulation follow a Boltzmann distribution.

Recall that we model the energy $E(x)$ as $E(x; \theta)$, and the force can be calculated as $F(x; \theta) = \nabla E(x; \theta)$. It follows from Hyvärinen & Dayan (2005) that we can learn the score function of the Boltzmann distribution using score matching, where the score function is defined as the gradient of the log density function $\nabla_x \log p(x)$. In our case, we observe that the force on a configuration $x$ is proportional to the score function, i.e., $F(x) \propto \nabla_x \log p(x)$. Therefore, we can use score matching to learn the forces by minimizing the unsupervised loss

$$L(\theta) = \mathbb{E}_{p(x)}\left[\frac{1}{k_\beta T}\text{Tr}[\nabla_x F(x; \theta)] + \frac{1}{2}||F(x; \theta)||^2\right]. \tag{3}$$

Although this allows us to solve the force matching problem in an unsupervised manner, the unsupervised loss is difficult to optimize in practice. To reduce the cost of solving Eq. 3, we adopt sliced score matching (Song et al., 2020). Sliced score matching takes advantage of random projections to significantly reduce the cost of solving Eq. 3, allowing us to apply score matching to large neural models such as GemNet.

In our experiments, we find that score matching does not match the accuracy of CCSD(T) force labels. Instead, we can think of score-matching as a form of inaccurate training. We therefore use score matching as an alternative to stages one and two of the ASTEROID framework. That is, we minimize Eq. 3 to get $\theta_{\text{init}}$, after which the model is fine-tuned on the accurate data.

## 5 EXPERIMENTS

### 5.1 DATA AND MODELS

For our main experiments, we consider three settings: using DFT data to enhance CCSD(T) training, using empirical force field data to enhance CCSD(T) training, and using unlabelled configurations to enhance CCSD(T) training. For the CCSD(T) data, we use MD17@CCSD, which contains 1,000 configurations labelled at the CCSD(T) and CCSD level of accuracy for 5 molecules (Chmiela et al., 2017). For DFT data, we use the MD17 dataset which contains over 90,000 configurations labelled at the DFT level of accuracy for 10 molecules (Chmiela et al., 2017). For the empirical force field data, we generate 100,000 configurations using the OpenMM empirical force field software (Eastman et al., 2017). For the unlabelled datasets, we use MD17 with the force labels removed.

In each setting we use 200, 400, 600, 800, or 950 CCSD(T) training samples as accurately labelled data. A validation set of size 50 and a test set of size 500 is used in all experiments, except for ethanol, where a test set of size 1000 is used. Recall that inaccurately labelled datasets can be generated at a tiny of the cost of CCSD(T) data.

We implement our method on GemNet and EGNN. For GemNet we use the same model parameters as Gasteiger et al. (2021). For EGNN, we use a 5-layer model and an embedding size of 128. We also compare ASTEROID to sGDML (Chmiela et al., 2019), a kernel-based method that has been shown to perform well when limited training data is available. For training with inaccurate data, we train with a batch size of 16 and stop training when the validation loss stabilizes. In the fine-tuning stage, we use a batch size of 10 and train for a maximum of 2000 epochs. To tune the bias aware loss parameter $\gamma$, we search in the set $\{0.5, 0.75, 1.0, 2.0, 3.0, 4.0, 5.0\}$ and select the model with the lowest validation loss. Comprehensive experimental details are deferred to Appendix A.1.

## 5.2 ENHANCING FORCE FIELDS WITH DFT

We display the results for using DFT data to enhance CCSD(T) training in Figure 2 for GemNet and Figure 3 for EGNN. From these figures, we can see that ASTEROID can outperform standard training for all amounts of data. In particular, the performance gain from using ASTEROID is strongest when limited amounts of accurate data are available and decreases as the amount of accurate data increases. When applied to GemNet in low resource settings, ASTEROID on average improves prediction accuracy by 52% and sample efficiency by a factor of 2.4. For EGNN, ASTEROID improves prediction accuracy by 66% and increases sample efficiency by more than 5 times. The large performance increase for EGNN may be due to the fact that the EGNN architecture has less inductive bias than GemNet, and therefore may struggle to learn the structures of the underlying force field with only a small amount of data.

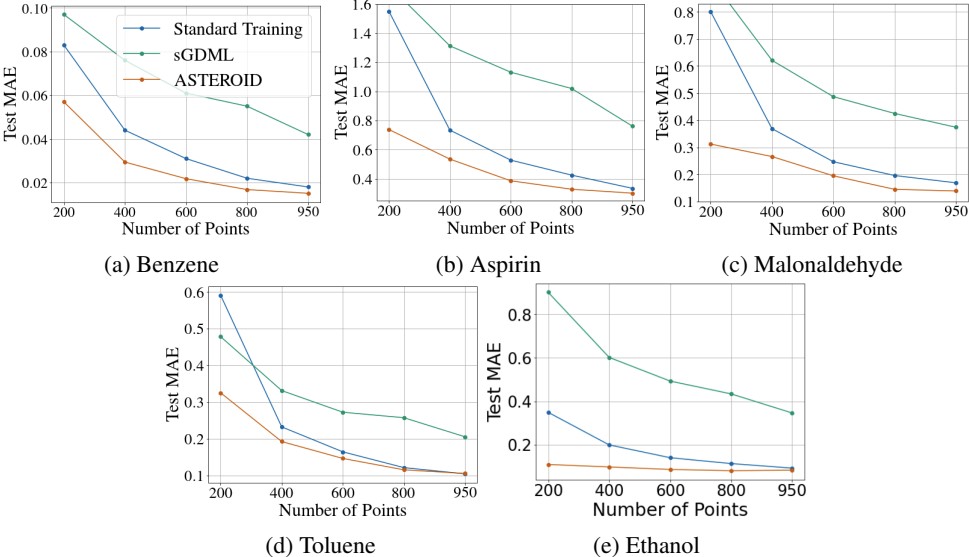

Figure 2: Main results for GemNet when DFT data is viewed as inaccurate.

## 5.3 ENHANCING FORCE FIELDS WITH EMPIRICAL FORCE CALCULATION

We present the results for empirical force field in Table 1. Due to limited space, we only display the results for the case where 200 accurate data points are available. The remaining results for GemNet can be found in Appendix A.2. Again we find that ASTEROID significantly outperforms the supervised baseline, improving prediction accuracy by 36% for GemNet and by 17% for EGNN. We note that empirical force fields are typically much less accurate than DFT (see Figure 4) but also much faster to compute. Despite this accuracy gap, the ASTEROID framework is able to utilize the empirical force field data to improve performance, which allows ASTEROID to benefit from even cheaper labels than DFT and obtain better efficiency.

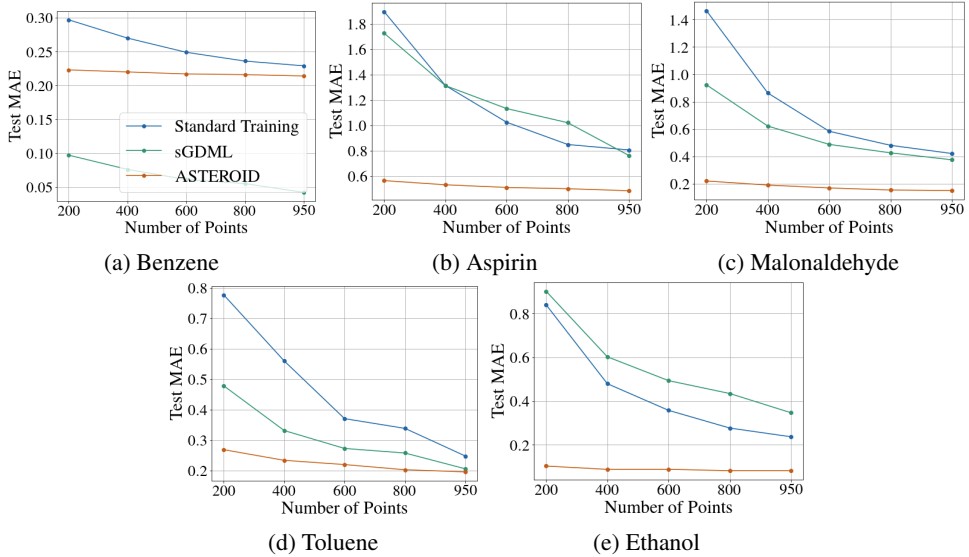

Figure 3: Main results for EGNN when DFT data is viewed as inaccurate.

Table 1: Accuracy of ASTEROID with empirical force field data. The training set for the fine-tuning stage contains 200 CCSD(T) labelled molecules.

|  | Aspirin | Benzene | Malonaldehyde | Toluene | Ethanol |
|---|---|---|---|---|---|
| **GemNet** | | | | | |
| Standard Training | 1.554 | 0.083 | 0.801 | 0.591 | 0.348 |
| ASTEROID | **0.843** | **0.048** | **0.516** | **0.337** | **0.301** |
| **EGNN** | | | | | |
| Standard Training | 1.897 | 0.297 | 1.466 | 0.777 | 0.840 |
| ASTEROID | **1.314** | **0.268** | **1.341** | **0.664** | **0.637** |

## 5.4 ENHANCING FORCE FIELDS WITH UNLABELLED MOLECULES

We first verify that score matching can learn the forces on unlabelled molecules by comparing its prediction accuracy on CCSD(T) data with models trained on supervised data (DFT and empirical force fields). We show the results in Figure 4. Surprisingly, we find that the prediction error of score matching is between that of DFT and empirical force fields. This indicates that relatively accurate force predictions can be obtained only by solving Eq. 3. Next we extend ASTEROID to settings where unlabelled data is available by fine-tuning the model obtained from score matching. We present the results for using ASTEROID on unsupervised configurations in Table 2, where we find that ASTEROID can improve prediction accuracy by 18% for GemNet and 4% for EGNN.

## 5.5 ABLATION STUDIES

We study the effectiveness of each component of ASTEROID. Specifically, we investigate the importance of bias-aware training (BAT) and fine-tuning (FT) when compared with standard training. As can be seen in Table 3, each of ASTEROID's components is effective and complementary to one another. We find that bias-aware training is most helpful with GemNet, which has more capability to overfit harmful data points than EGNN.

## 5.6 ANALYSIS

⋄ **Size of inaccurate data.** To demonstrate that ASTEROID can exploit varying amounts of inaccurate data, we plot the performance of ASTEROID with randomly sub-sampled inaccurately

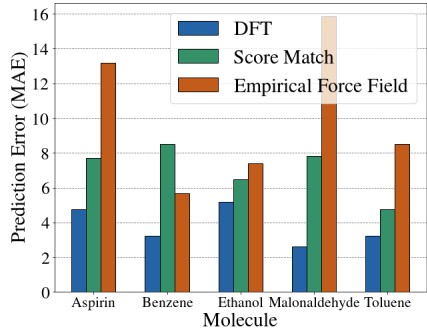

Figure 4: Prediction errors of models tested on CCSD(T) data. Models are trained with different inaccurately labelled data but are not fine-tuned on the accurate CCSD(T) data.

Table 2: Accuracy of ASTEROID with unlabelled molecular configurations. The training set for the fine-tuning stage contains 200 CCSD(T) labelled molecules.

|  | Aspirin | Benzene | Malonaldehyde | Toluene | Ethanol |
|---|---|---|---|---|---|
| **GemNet** | | | | | |
| Standard Training | 1.554 | **0.083** | 0.801 | 0.591 | 0.348 |
| ASTEROID (Unsupervised) | **0.928** | 0.093 | **0.629** | **0.475** | **0.314** |
| **EGNN** | | | | | |
| Standard Training | 1.897 | **0.297** | 1.466 | 0.777 | 0.840 |
| ASTEROID (Unsupervised) | **1.756** | 0.305 | **1.382** | **0.740** | **0.823** |

Table 3: Ablation study on ASTEROID. The inaccurate data is DFT labelled configurations and the accurate dataset contains 200 CCSD(T) labelled configurations.

|  | Aspirin | Benzene | Malonaldehyde | Toluene | Ethanol |
|---|---|---|---|---|---|
| **GemNet** | | | | | |
| Standard Training | 1.554 | 0.083 | 0.801 | 0.591 | 0.348 |
| ASTEROID w/o FT | 4.670 | 3.252 | 2.726 | 3.342 | 5.107 |
| ASTEROID w/o BAT | 0.788 | 0.062 | **0.311** | 0.360 | 0.112 |
| ASTEROID | **0.738** | **0.057** | 0.312 | **0.325** | **0.109** |
| **EGNN** | | | | | |
| Standard Training | 1.897 | 0.297 | 1.466 | 0.777 | 0.840 |
| ASTEROID w/o FT | 4.617 | 3.260 | 2.765 | 3.304 | 5.120 |
| ASTEROID w/o BAT | 0.577 | 0.225 | **0.213** | 0.272 | **0.009** |
| ASTEROID | **0.563** | **0.223** | 0.219 | **0.268** | 0.102 |

labelled data. ASTEROID performs best when large amounts of inaccurate data are available, but still increases the accuracy by 32% when trained using small amounts of inaccurately labelled data.

We also investigate the performance of score matching when varying amounts of unlabelled data are available. We find that score matching is fairly robust to the number of available configurations, beating empirical force fields with only 100 configurations and only requiring 5000 configurations for optimal performance.

◇ **Runtime Comparison.** Although there is no official documentaion on the time needed to create the DFT and CCSD(T) data in the MD17 dataset, we do create our own empirical force field dataset and measure the time needed to create it. Generating 100,000 force molecular conformations and their respective force labels takes roughly two hours with a single V100 32GB GPU. This is much

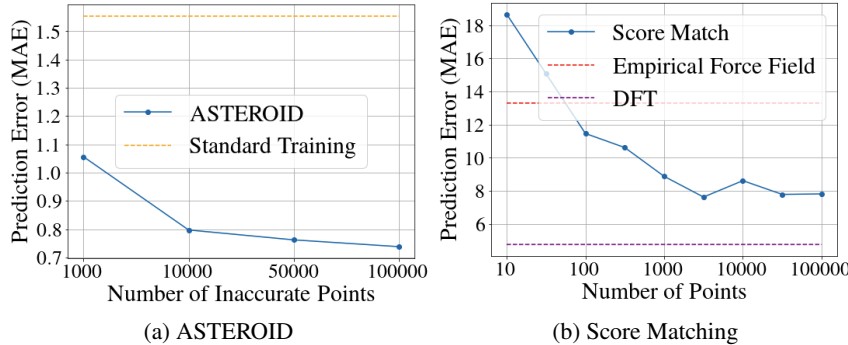

(a) ASTEROID  (b) Score Matching

Figure 5: Prediction error of models trained on with varying amounts of inaccurate data. The molecule is aspirin, the inaccurate data is labelled with DFT, and the accurate data is made up of 200 CCSD(T) labelled molecules. For score matching we do not fine-tune the model.

faster than DFT or CCSD(T), which can take hundreds or thousands of seconds for a single computation (Bhattacharya et al., 2021; Datta & Gordon, 2021). In general, it is difficult to get an exact time comparison between DFT and CCSD(T) because the runtime depends on the molecule, the number of basis functions used, and the implementation of each algorithm.

◇ **Hyperparameter Sensitivity.** We investigate the sensitivity of ASTEROID to the hyperparameters $\gamma$ and $\rho$ (note that $\rho$ is only used in GemNet). From the Figure 6 we can see that ASTEROID is robust to the choice of hyperparameters, outperforming standard training in every setting.

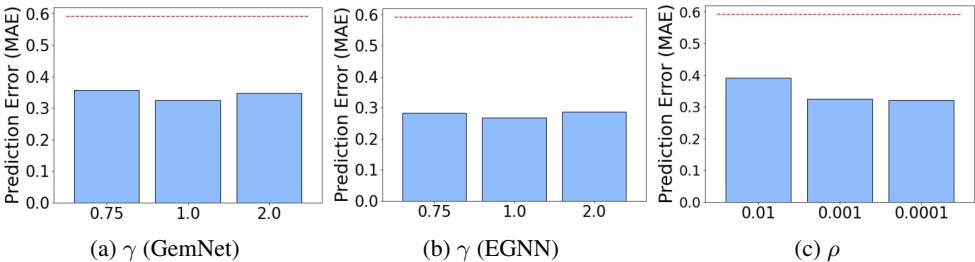

(a) $\gamma$ (GemNet)  (b) $\gamma$ (EGNN)  (c) $\rho$

Figure 6: Ablation study on the toluene molecule. We set $\gamma = 1.0$ and $\rho = 0.001$ by default. The red line represents standard training.

## 6  DISCUSSION AND CONCLUSION

One related area of work is $\Delta$-machine learning (Ramakrishnan et al., 2015; Ruth et al., 2022; Nandi et al., 2021), which also uses both inaccurate force calculations and accurate force calculations. Specifically, $\Delta$-machine learning learns the difference between inaccurate and accurate force predictions, therefore speeding up MD simulation while maintaining high accuracy. However, such an approach requires an equal amount of accurate and inaccurate training data, which therefore makes the training data expensive to generate. Another related area of work is training general purpose MLFFs over many molecules (Smith et al., 2019; Deringer et al., 2020). Different from our work, these methods require lots of data from multiple molecules and train on a huge dataset that is not generated in a cost-aware manner. Additionally, the method from Smith et al. (2017) only trains on equilibrium states and may not work well for MD trajectory data. Numerical comparisons can be found in Appendix A.7.

Different from previous works on machine learning force fields, we propose to learn MLFFs in a data cost aware manner. Specifically, we propose a new training framework, ASTEROID, to boost prediction accuracy with cheap inaccurate data. In addition, we extend ASTEROID to the unsupervised setting. By learning force field structures on the cheap data, ASTEROID only requires a small amount of expensive and accurate datapoints to achieve good performance. Extensive experiments validate the efficacy of ASTEROID.

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

## A  APPENDIX

### A.1  EXPERIMENTAL DETAILS

In this section we go over the experimental details.

**GemNet Training Details.** To train the bias identification method, we train a freshly initialized model with a batch size of 10 on the accurate dataset for 2000 epochs. To train the inaccurate model, we train a freshly initialized model with the bias aware loss function and batch size 16 over the inaccurate dataset. Finally to finetune the inaccurately trained model, we train a model with a batch size of 10 on the accurate dataset for 2000 epochs. In each stage of training we use the following hyperparamers:

- Evaluation Interval: 1 epoch

- Decay steps: 1200000

- Warmup steps: 10000

- Decay patience: 50000

- Decay cooldown: 50000

The rest of the parameters are the same as used in Gasteiger et al. (2021).

**EGNN Training Details.** The EGNN training setup is similar to GemNet. To train the bias identification method, we train a freshly initialized model with a batch size of 10 on the accurate dataset for 2000 epochs. To train the inaccurate model, we train a freshly initialized model with the bias aware loss function and batch size 32 over the inaccurate dataset. Finally to finetune the inaccurately trained model, we train a with a batch size of 10 on the accurate dataset for 2000 epochs. In each stage of training we use the following hyperparamers:

- Evaluation Interval: 1 epoch

- Learning rate: $10^{-4}$ for inaccurate training, $10^{-5}$ for finetuning

- num_layers: 5

- embedding_size: 128

### A.2  ADDITIONAL RESULTS

Here we include additional results for ASTEROID when empirical force field data is viewed as inaccurate. For the baseline model we use GemNet. The ASTEROID framework again leads to consistent gains across all amounts of data.

### A.3  DERIVATION OF SCORE MATCHING FOR FORCES

For a given molecule with conformations $x_1, .., x_n$, let us denote energy as $E(x)$. Then the the Boltzmann/Equilibrium distribution for the molecule is given by

$$p(x) = \frac{1}{Z}\exp(-\beta E(x)),$$

where $Z$ is a normalizing constant, $\beta = \frac{1}{k_\beta T}$, $k_\beta$ is the Boltzmann constant, and $T$ is the temperature under which the simulation is run. Then we can see that the force on a conformation $x$ is equivalent to the score, i.e. $F(x) = -\nabla_x E(x) = \frac{1}{\beta}\nabla_x \log p(x)$. Therefore learning the force $F(x)$ is equivalent to learning the score $\frac{1}{\beta}\nabla_x \log p(x)$. Suppose we parameterize the MLFF to directly predict the force as $F_\theta(x)$. Then the force matching loss can be written as

$$\mathcal{L}(\theta) = \frac{1}{2}E_{x\sim p(x)}\|F_\theta(x)-F(x)\|_2^2 = \frac{1}{2}E_{x\sim p(x)}\|F(x)\|_2^2 - E_{x\sim p(x)}\left[\langle F_\theta(x), F(x)\rangle\right] + \frac{1}{2}E_{x\sim p(x)}\|F_\theta(x)\|_2^2.$$

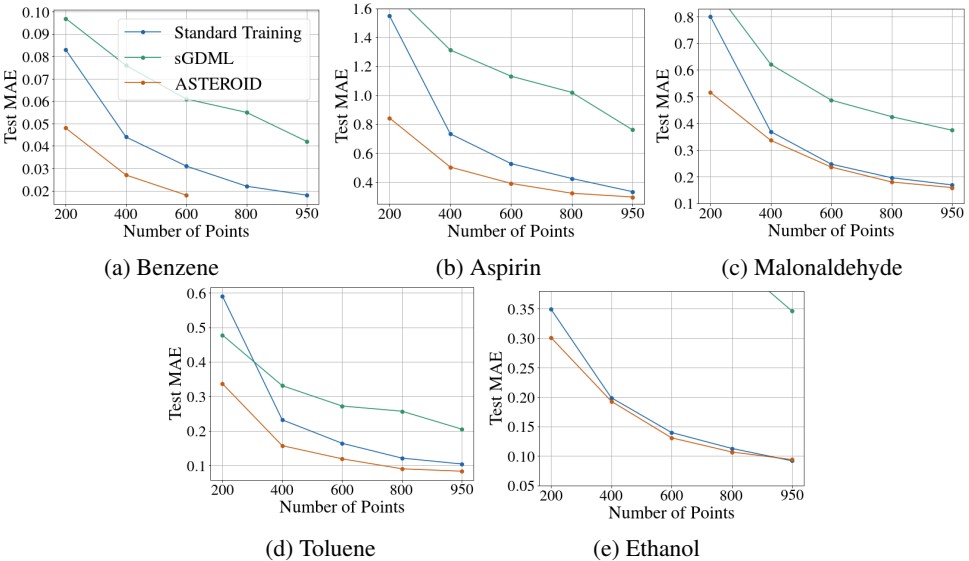

Figure 7: Main results for GemNet when DFT data is viewed as inaccurate.

The middle term can then be expanded as

$$
\begin{aligned}
E_{x \sim p(x)}\left[\langle F_\theta(x), F(x)\rangle\right] &= \int_x p(x)\langle F_\theta(x), F(x)\rangle dx \\
&= \int_x p(x) \sum_{i=1}^{d}\left(\frac{1}{\beta}\frac{d\log p(x)}{dx_i}F_\theta(x)_i\right)dx \\
&= \frac{1}{\beta}\sum_{i=1}^{d}\int_x \frac{d\log p(x)}{dx_i}F_\theta(x)_i dx \\
&= \frac{1}{\beta}\sum_{i=1}^{d}\int_{x_{i^-}}\int_{x_i} F_\theta(x)_i dp(x)d_{x_{i^-}} \\
&= \frac{1}{\beta}\sum_{i=1}^{d}\int_{x_{i^-}}\left(F_\theta(x)_i dp(x)|_{-\infty}^{+\infty} - \int_{x_i}p(x)dF_\theta(x)_i\right)d_{x_{i^-}} \\
&= -\frac{1}{\beta}\sum_{i=1}^{d}\int_{x_{i^-}}\int_{x_i}p(x)\frac{dF_\theta(x)_i}{dx_i}dx_i d_{x_{i^-}} \\
&= -\frac{1}{\beta}\sum_{i=1}^{d}E_{x \sim p(x)}\left[\frac{dF_\theta(x)_i}{dx_i}\right] = -\frac{1}{\beta}E_{x \sim p(x)}\left[\mathrm{Tr}\left[\nabla_x F_\theta(x)\right]\right].
\end{aligned}
$$

Therefore we have the loss

$$
\mathcal{L}(\theta) = E_{x \sim p(x)}\left[\frac{1}{\beta}\mathrm{Tr}\left[\nabla_x F_\theta(x)\right] + \frac{1}{2}\|F_\theta(x)\|_2^2\right].
$$

### A.4 ASTEROID DIAGRAM

In order to to make the ASTEROID framework more clear, we provide a workflow diagram.

### A.5 ASTEROID TOY EXAMPLE

We have added a new result using a two-layer MLP with 128 hidden units each and synthetic data. This experiment shows that ASTEROID can significantly improve generalization error in a variety of

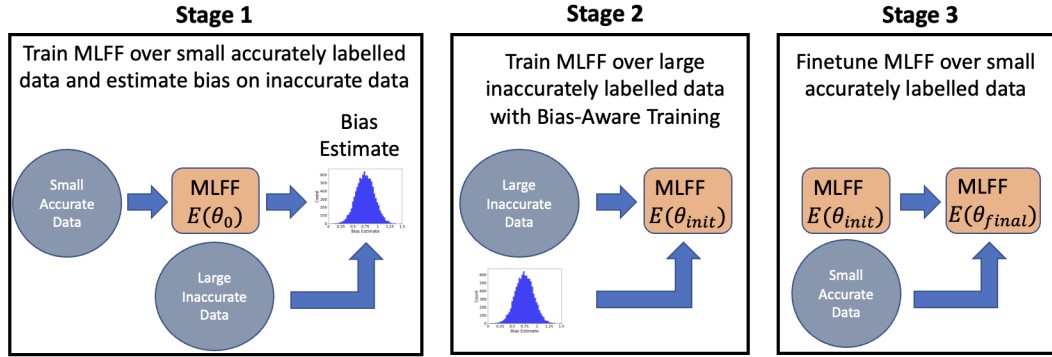

Figure 8: Asteroid workflow diagram.

settings. In this experiment, we generate a biased dataset of 2000 points according to $Y = AX + b$, where where $X \sim N(0, 1)$ has dimension 16, $b$ is the bias, and $A$ is a randomly generated Gaussian matrix of dimension $16 \times 16$. The bias b is chosen uniformly from the set $[0, 2, 4, 8, 16]$. We also generate varying levels of accurate data according to $Y = AX$, where $X \sim N(0, 1)$. We then evaluate the test MAE of ASTEROID and standard training over a variety of accurate data sizes. We find that ASTEROID significantly outperforms standard training.

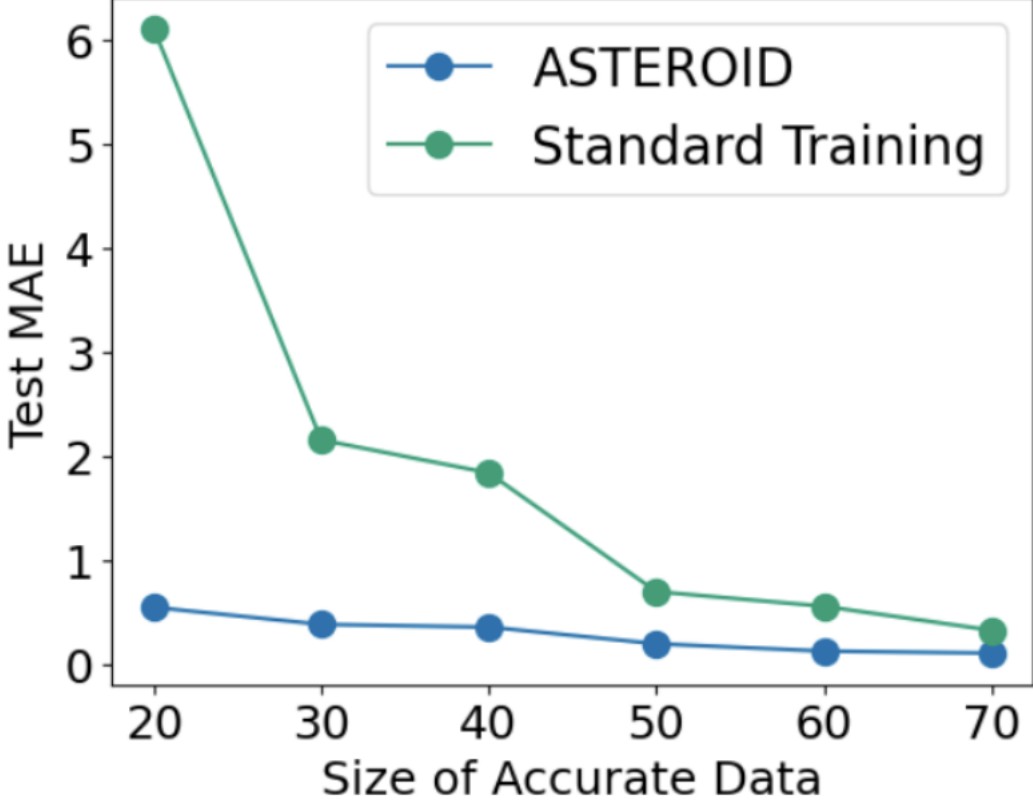

Figure 9: Asteroid toy example.

## A.6 MOLECULAR DYNAMICS SIMULATION

We also evaluate the performance of MLFFs trained by ASTEROID in downstream MD simulation tasks. First, we demonstrate that ASTEROID trained MLFFs can produce stable dynamics, while

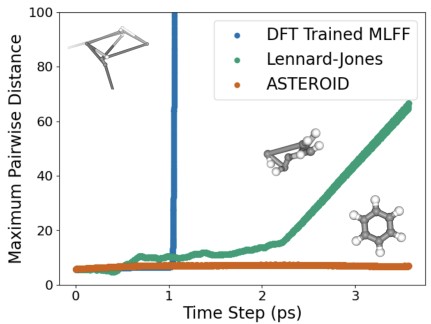
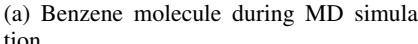
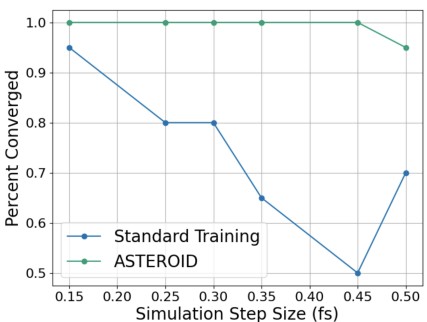

(a) Benzene molecule during MD simulation.

(b) Proportion of converged simulations (Aspirin).

both MLFFs trained on on DFT data alone and empirical force field diverge. Using the Atomic Simulation Environment (ASE) (Larsen et al., 2017), we simulate the behavior of a Benzene molecule using forces calculated by a MLFF trained with ASTEROID, a MLFF trained on DFT data only, and the Lennard-Jones empirical force field. We simulate the molecule with Langevin dynamics, where the steps size is 0.5 femtoseconds, the temperature is 500K, the friction coefficient is 0.002, and the maximum number of time steps is 10000. The results of these simulations can be seen in 10a, where ASTEROID is able to produce stable dynamics, while the error compounding effect of the DFT trained force field and the Lennard-Jones potential is too heavy, resulting in a diverged simulation.

Next we compare the simulation ability of ASTEROID with that of standard training, where both MLFFs are trained on 200 points of CCSD(T) data. To evaluate the robustness of the MLFFs, we run MD simulations with varying step sizes on the Aspirin molecule. The setting is the same as in Figure 10a. In Figure 10b we plot the number of simulations that do not converge out of 20 random runs. The ASTEROID framework is able maintain steady performance across step sizes and almost all the simulations converge. In contrast, the simulations powered by standard MLFFs fail with larger step size.

### A.7 COMPARISON WITH ANI-1CCX

We compare the performance of the method from Smith et al. (2017) with ours in Table 4, in both the case when the ANI-1 model is fine tuned on MD17 and the zero-shot setting. Despite having a much larger data generation budget, the open-domain pre-training of Smith et al. (2017) is not as powerful as ASTEROID.

Table 4: Accuracy of ASTEROID compared with ANI-1ccx. The training set for the fine-tuning stage contains 200 CCSD(T) labelled molecules. FT refers to fine-tuning ANI-1ccx on MD17.

|                     | Aspirin | Benzene | Malonaldehyde | Toluene | Ethanol |
|---------------------|---------|---------|---------------|---------|---------|
| **ASTEROID (GemNet)** | 0.843 | 0.048 | 0.516 | 0.337 | 0.301 |
| **ANI-1ccx** | 1.897 | 0.297 | 1.466 | 0.777 | 0.840 |
| **ANI-1ccx (FT)** | 1.314 | 0.268 | 1.341 | 0.664 | 0.637 |

