# OpenReview forum: "Machine Learning Force Fields with Data Cost Aware Training"
_ICLR.cc/2023/Conference — Submitted to ICLR 2023_

### Official Review · Reviewer_Y3t9 · 2022-10-22

**Confidence:** 4
**Correctness:** 3
**Technical Novelty And Significance:** 2
**Empirical Novelty And Significance:** 2
**Recommendation:** 3

**Clarity, Quality, Novelty And Reproducibility:**

Clarity:
Although there is an exception, as I asked above, most of the paper is well-written.

Quality:
My main concern is that the evaluation is only about the accuracy of its force field using a small dataset.  However, it is important to convince readers about the performance of the proposed method in real-world applications.

Novelty:
The proposed method is a strait-forward extension of previous work, so the technical contribution is limited.

Reproducibility:
Although there is an exception, as I asked above, most of the experiment settings are clear for reproduction.


**Details Of Ethics Concerns:**

No concern

**Strength And Weaknesses:**

Strength
* It is interesting to see the pretraining results using a dataset collected by not only DFT but also the empirical force field.
* The use of score-matching for unsupervised pretraining is reasonable approach.
* Unsupervised pretraining of NNP might be useful since it might be possible to aggregate multiple data sets collected by different methods.
The proposed method down-weight structures with higher energy, but we also need the energy/force estimation for disordered structures, i.e., structures of higher energy, in real-world applications, such as MD simulations or structural optimization.

Weaknesses
* There is no evaluation result on its applications, such as MD simulations or structural optimization.  Since force accuracy is not sufficient for obtaining stable dynamics [1], I am not confident about the effectiveness of the proposed approach for its applications.
* I think ANI- 1ccx  (Smith et al., 2019), a simple pretraining-finetuning method, should be a baseline method.  In my understanding, ASTEROID w/o BAT is the proposed transfer learning in Smith et al., 2019.  However, the results of it are not available in Figures 2, 3, 4, and Table 2.
* If the unsupervised learning method is used for the second stage, the first stage is meaningless.  Thus, the unsupervised learning method does not extend ASTEROID in this case, and this contradicts the section title of Section 4.   In addition, the experiments conducted in this case.


Questions:
1.  Can you add the simple pretraining-finetuning results, i.e., ASTEROID w/o BAT, in Figures 2, 3, 4, and Table 2?   Since ANI- 1ccx  (Smith et al., 2019) is a suitable baseline method, I think you can omit "Standard Training" results if there is no space available.

2.  I miss the meaning of "Next we extend ASTEROID to settings where unlabelled data is available by fine-tuning the model obtained from score matching".
Is this mean that the unlabeled data is collected by MD simulations using a fine-tuned model?  Or does it simply mean you pre-trained an NNP using unlabelled data and fine-tuned it using CCSD(T) data?

3.  Can you provide any results of applications for which DFT-based NNP fails, as Smith et al., 2019 did?


[1] Sina Stocker, Johannes Gasteiger, Florian Becker, Stephan Günnemann, and Johannes Margraf. How robust are modern graph neural network potentials in long and hot molecular dynamics simulations?  2022.


**Summary Of The Paper:**

This paper address NNP training using an expensive data generation method, CCSD(T).
The proposed method is based on the pretraining of NNP using the training data collected by relatively low-cost data generation methods, such as DFT.
The authors extended it to a three-stage method, in which an importance-weighting method is used for the pretraining of NNP.
An NNP is trained using CCSD(T) at the first stage, and the instance weights for the pretraining are estimated by the discrepancy between the force of low-cost data generation methods and the force of the NNP trained using CCSD(T).   In addition, the authors propose unsupervised pretraining using score matching for the pretraining.
The proposed method outperforms the baseline methods on the MD17 dataset.

**Summary Of The Review:**

The fatal problem of the paper is no evaluation result on its applications, such as MD simulations or structural optimization.
In addition, the baseline algorithm in the experiments is also not appropriate.
I think a major revision is required for acceptance.

---

> ### Author Response · Authors · 2022-11-19
> **Thank you for the feedback. We have added MD simulation results.**
>
> We appreciate your constructive suggestions. We provide our response to the main comments as follows.
>
> **Quality: My main concern is that the evaluation is only about the accuracy of its force field using a small dataset. However, it is important to convince readers about the performance of the proposed method in real-world applications:**
>
> Based on your suggestions, we have added experiments to evaluate the trained MLFF model on downstream MD simulation tasks. As mentioned before, we find that in contrast to other methods such as empirical force fields, ASTEROID is able to produce stable dynamics. In addition, we find that the MLFFs trained by ASTEROID are more robust than those trained via standard methods, due to its increased accuracy and the fact that ASTEROID training is augmented by many cheaply available conformations. A more detailed description of these experiments can be found in appendix A.6.
>
> **I think ANI- 1ccx (Smith et al., 2019), a simple pretraining-finetuning method, should be a baseline method. In my understanding, ASTEROID w/o BAT is the proposed transfer learning in Smith et al., 2019. However, the results of it are not available in Figures 2, 3, 4, and Table 2:**
>
> Thank you for the suggestion! We note that different from our method, ANI-1ccx conducts open domain pre-training over a large set of molecules, with the goal of being a general purpose MLFF. We have evaluated ANI-1ccx on our datasets in both the zero-shot setting and with fine tuning. The results are added to Table 4 in appendix A.7. Although ANI-1ccx was trained on over 20-million molecular conformations, our method is much more accurate while requiring significantly less and cheaper data. We hypothesize that training a unified force field model such as ANI-1ccx is challenging.
>
> **If the unsupervised learning method is used for the second stage, the first stage is meaningless. Thus, the unsupervised learning method does not extend ASTEROID in this case, and this contradicts the section title of Section 4.:**
>
> With unsupervised pre-training, we do not use a bias-aware loss function. Score-matching does not necessarily extend ASTEROID, but can be seen as a variant following the general formula of training on noisy data and then fine tuning on more expensive data. We have made the clarification in the revision.
>
> **I miss the meaning of "Next we extend ASTEROID to settings where unlabelled data is available by fine-tuning the model obtained from score matching". Is this mean that the unlabeled data is collected by MD simulations using a fine-tuned model? Or does it simply mean you pre-trained an NNP using unlabelled data and fine-tuned it using CCSD(T) data?**
>
> In this case, we pre-train the NNP using score-matching, and then fine-tune on CCSD(T) data. We have made clarification in the revision.
>
> **Can you provide any results of applications for which DFT-based NNP fails, as Smith et al., 2019 did?**
>
> Yes. We find that when used in MD-simulation, the models trained with DFT data only diverge, while the models trained with ASTEROID can produce stable trajectories. More details can be found in appendix A.6.

---

### Official Review · Reviewer_VP9R · 2022-10-25

**Confidence:** 2
**Correctness:** 3
**Technical Novelty And Significance:** 3
**Empirical Novelty And Significance:** 2
**Recommendation:** 6

**Clarity, Quality, Novelty And Reproducibility:**

Clarity & Quality: Overall the clarity and quality of the work is high in terms of outline but description of datasets is limited and need more explanation. Some graphing could be better eg. Ethanol plotting for GemNet is out of the charts. Also metrics in some tables is missing (eg. what exactly are we measuring in table 1? The paper says accuracy but needs to be part of the table legend for additional clarity). Given this a complex area, a workflow diagram in introduction or shortly after that walks through the key steps from end to end (similar to an architecture diagram) can help simplify understanding.


Novelty: Overall the method is reasonably novel in terms of an alternative mixture of training strategies although the individual steps themselves aren't novel. Key comparisons in terms of time effort over other methods can demonstrate additional novelty.


Reproducibility:
1. ASTEROID implementation or algorithm isn't provided under references.
2. Calculation and analysis of experimental procedures is unavailable for reproducibility.

**Strength And Weaknesses:**


Strength:
1. Several performance comparisons around accuracy are method with different methods (state of the art) under various conditions to really drive home the point around performance improvement and superiority over existing methods
2. Sufficient background provided to identify gaps and opportunities for development, although there could be more clarity around datasets and overall learning workflows.


Weaknesses:
1. Description of datasets is complex, and not simplified eg. what does accurate, inaccurate data look like along with labels.
2. Method to estimate bias, metrics around bias measurement/identification could be clearer, eg. why is there a surrogate for bias measurement and how would ideal bias measurement look like prior to introducing surrogate method.
3. Need additional commentary on any limitations of this method eg. cases where such an approach would not work and what could be alternatives.
4. Calculations/stats/graphs around time comparison with other methods is not available which could be a key result.
5. No clear explanation why the 5 particular chemicals were chosen.
6. Also why test size is different for ethanol vs others? Do other molecules also need to be compared with the same test set size?
7. For the hyper parameter tuning setting, I dont see the comparison between ASTEROID and standard training, even though its mentioned that way.

**Summary Of The Paper:**

The authors in this paper, propose a framework which enables training data generation with is computationally inexpensive while being accurate for molecular dynamics simulation. The authors argue that the current state of the art for machine learning force fields are computationally very expensive when it comes to achieving high accuracy, including generating a large amount of costly training data. While there are options to reduce computational complexity by training on fewer data points or cheaper reference forces, these options come with a tradeoff of loss in accuracy/poor performances.

The authors propose that ASTEROID works in a multi stage training framework where step 1: small amounts of accurate data is first identified in a large inaccurate dataset, step 2: train the models on large amounts of training data which is inaccurate and cheaply available, followed by step 3: where small amounts of accurate training data which is more expensive to obtain. step 1 is achieved via a bias aware loss function to prevent overfitting on inaccurate training data. The authors demonstrate that such a method significantly outperforms conventional methods such as standard training and sGDML on networks such as GEMNet and EGNN


**Summary Of The Review:**

I propose this work is conditionally accepted with few more experiments such as time component comparison with other methods to demonstrate cost awareness and efficiencies to drive home the point of why the method is superior over prior work of art. While the authors argue that the method is superior in terms of accuracy over other current work, additional metrics are needed to really drive home the point without which the work seems less convincing. Also even though there is no availability of materials to test reproducibility, it is expected that a workflow to guide experimental setup and analysis would be necessary to validate some of the results.

---

> ### Author Response · Authors · 2022-11-19
> **Thank you for the helpful feedback.**
>
> **Description of datasets is complex, and not simplified eg. what does accurate, inaccurate data look like along with labels:**
>
> Note that the difference between the inaccurate data and accurate data comes from the level of accuracy of the force label, and not from the molecular conformations themselves. Ideally, we would like to predict the ground truth force (denoted as $Y_{g}$) on a molecular conformation. The inaccurate data are molecular conformations with biased force labels, i.e. $Y_I = Y_{g} + $bias. The accurate data are molecular conformations with labels very similar to the ground truth labels.
>
> **Method to estimate bias, metrics around bias measurement/identification could be clearer, eg. why is there a surrogate for bias measurement and how would ideal bias measurement look like prior to introducing surrogate method.:**
>
> An ideal bias measurement for an inaccurate force label would mean calculating the difference between the inaccurate force label and the ground truth label. However, we do not have access to ground truth labels on the inaccurate dataset. Therefore we estimate the ground truth labels with a model trained on accurate data. The surrogate for bias is then calculated as the distance between the inaccurate force label and the estimated ground truth label. With ASTEROID, we assume that the variance contributes similarly to the label error, so a larger distance between the inaccurate force label and the estimated ground truth label implies a larger bias.
>
> **Need additional commentary on any limitations of this method eg. cases where such an approach would not work and what could be alternatives:**
>
> This field is very new, so we are not aware of immediate limitations. One possible limitation is that the additional pre-training stage will require more computational resources. This was not a problem in our experiments, as the computational cost of pre-training is comparable to that of fine tuning. In general for modern MLFF models the training costs are still much lower than those required to generate accurate force labels.
>
> **Calculations/stats/graphs around time comparison with other methods is not available which could be a key result.:**
>
> The computation time of DFT and empirical force fields is negligible compared to CCSD(T)’s O($n^7$) scaling. In practice, the runtime of CCSD(T) and DFT depends on the software used and the choice of basis functions. We remark that there is no official documentation on the MD17 dataset runtime.
>
> We do generate our own dataset with empirical force fields, and find that 100,000 force computations can be done in less than two hours, while still significantly boosting performance.
>
> In addition, we note that molecular configurations cannot be generated with CCSD(T) force calculations, since this would take too long (over 100,000 force calculations can be needed to reach equilibrium). Instead, force calculations are done with DFT or empirical force fields to generate conformations. A subset of these points are then labeled with CCSD(T) calculations. Therefore, the inaccurate labels can typically be used with **no additional labeling cost**. We have added a note on runtime comparison to section 5.6.
>
> **No clear explanation why the 5 particular chemicals were chosen.:**
>
> These five molecules are part of the MD17 dataset, which is a commonly used benchmark for testing MLFFs. In addition, these molecules are very important for computational chemistry, as they form the fundamental building blocks for more complex molecules. Another useful quality of these datasets is that we can obtain both CCSD(T) force labels and MD17 force labels for these datasets, whereas other datasets may not have CCSD(T) force labels. Due to high computational costs, CCSD(T) datasets are limited.
>
> **Also why test size is different for ethanol vs others? Do other molecules also need to be compared with the same test set size?:**
>
> We follow the practice of GemNet and other previous papers that use this test set size for ethanol.
>
> **For the hyper parameter tuning setting, I dont see the comparison between ASTEROID and standard training, even though its mentioned that way:**
>
> We have added a comparison to standard training in the hyperparameter study section.
>
> **Clarity & Quality: Overall the clarity and quality of the work is high in terms of outline but description of datasets is limited and need more explanation. Some graphing could be better. Also metrics in some tables is missing. Given this a complex area, a workflow diagram in introduction or shortly after that walks through the key steps from end to end can help simplify understanding.**
>
> Thank you for the suggestions! We have fixed the charts and added more detailed descriptions to the tables. In addition, we have added a workflow diagram for more clarity (appendix A.4).
>
> The details on how to reproduce our experiments can be found in appendix A.1. In addition, we hope the workflow diagram can help with reproducibility.

---

### Official Review · Reviewer_2odL · 2022-10-25

**Confidence:** 3
**Correctness:** 3
**Technical Novelty And Significance:** 2
**Empirical Novelty And Significance:** 2
**Recommendation:** 3

**Clarity, Quality, Novelty And Reproducibility:**

The paper is clearly written. All experiments are documented in detail.

As stated above, there is a lack of novelty and empirical evaluation of the method.

**Strength And Weaknesses:**

### Strengths

The method is simple to implement. It allows using various kinds of additional data to improve the model's performance. The authors test their procedure thoroughly on five molecules and use different types of auxiliary data.

### Weaknesses

The procedure is basically limited to adding additional data to the training procedure, which is in itself not new. The novel idea is training a model to estimate the bias and reweight the data accordingly. However, this bias-aware training does not improve the model significantly, see Table 3. The weights based on the estimated bias are apparently very noisy, but they could be improved by updating them during training in a bootstrapping fashion.

The method has been tested empirically on several molecules, but the paper lacks theoretical validation that the method is sound. Moreover, it should be tested in a toy setting, such as doing regression with a few variables on generated data with varying noise levels.

Most importantly, the authors did only compare the performance of their models with respect to the number of accurate training samples. However, they should do so with respect to the overall runtime used to generate both the accurate and the inaccurate training data. Since the relative improvement decreases rapidly when increasing the number of accurate samples, it is not obvious whether it would not be better to just generate more accurate data.

**Summary Of The Paper:**

In the paper, the authors aim to improve methods to train machine learning force fields for MD simulations. These methods rely on accurate data, which is expensive to generate. They augment the training procedure with less accurate data being cheaper to generate. This is achieved by first training a model with a limited amount of accurate data, which is then used to estimate the bias of the less accurate data. The data is reweighted according to this bias, followed by training a model on those samples. In a third step, the model is fine-tuned on the accurate data.

The authors evaluate their method on five molecules. The learned force field seems more accurate than when using standard training, especially when using a small amount of accurate training data and performs similar or better than a related approach.

**Summary Of The Review:**

In conclusion, I favor to reject this paper. However, I am open for a discussion with the authors and the other reviewers.


Edit after rebuttal:

I appreciate the author's effort to improve their paper. The toy example they added is interesting. However, the setup seems counterintuitive to me. First, there should be noise being added to the data and its standard deviation could depend on the accuracy of the simulated data generation process. Second, I like the idea of adding a sampled bias, but why is it sampled uniformly from the values {0, 2, 4, 8, 16}? Thereby, 20% of the "biased" data is actually unbiased. I suggest drawing the bias from an interval such as [2, 6] or perhaps [0, 8] instead, limiting the number of accurate samples, or even using a constant bias with additional noise centered around 0.
Given that improved runtime is one of the major selling points of the paper, I think a runtime analysis is vital for the paper.

Therefore, I will not change my score, but encourage the authors to improve their draft and submit it to another venue.

---

> ### Author Response · Authors · 2022-11-19
> **Thank you for the constructive comments.**
>
> We appreciate your constructive suggestions. We provide our response to your questions as follows.
>
> **The procedure is basically limited to adding additional data to the training procedure, which is in itself not new:**
>
> Our main contribution is identifying an important problem that has not been considered in previous literature and proposing a methodology to solve it. Previous MLFF works do not take the data generation budget into consideration. This is not practical, and we propose to reduce the cost by using less accurate data. We do not claim pre-training using inaccurate data as our main contribution, it is just a tool we identify to reduce the data generation cost.
>
> Another contribution is that we propose to use score matching as a pre-training method. This has not been discussed in existing literature, and the topic may be of independent interest.
>
> **The method has been tested empirically on several molecules, but the paper lacks theoretical validation that the method is sound. Moreover, it should be tested in a toy setting, such as doing regression with a few variables on generated data with varying noise levels.:**
>
> Toy setting: We have added a new result using a two-layer MLP with 128 hidden units each and synthetic data. This experiment shows that ASTEROID can significantly improve generalization error in a variety of settings.
> In this experiment, we generate a biased dataset of 2000 points according to $Y = AX + b$, where where $X \sim N(0, 1)$ has dimension 16, $b$ is the bias, and $A$ is a randomly generated Gaussian matrix of dimension $16\times16$. The bias b is chosen uniformly from the set $[0, 2, 4, 8, 16]$. We also generate varying levels of accurate data according to $Y = AX$, where $X \sim N(0,1)$. We then evaluate the test MAE of ASTEROID and standard training over a variety of accurate data sizes. We find that ASTEROID significantly outperforms standard training. The results of this experiment can be seen in appendix A.5.
>
> **Most importantly, the authors did only compare the performance of their models with respect to the number of accurate training samples. However, they should do so with respect to the overall runtime used to generate both the accurate and the inaccurate training data.:**
>
> The computation time of DFT and empirical force fields is negligible compared to CCSD(T)’s $O(n^7)$ scaling. In practice, the runtime of CCSD(T) and DFT depends on the software used and the choice of basis functions. We remark that there is no official documentation on the MD17 dataset runtime.
>
> We do generate our own dataset with empirical force fields, and find that a dataset of 100,000 conformations can be generated in less than two hours, while still significantly boosting performance.
>
> In addition, we note that molecular configurations cannot be generated with CCSD(T) force calculations alone, since this would take too long (over 100,000 force calculations can be needed to reach equilibrium). Instead, force calculations are done with DFT or empirical force fields to generate conformations. A subset of these points are then labeled with CCSD(T) calculations. Therefore, the inaccurate labels can typically be used with **no additional labeling cost**. We have added a note on runtime comparison to section 5.6.

---

### Official Review · Reviewer_K6wT · 2022-10-31

**Confidence:** 4
**Correctness:** 3
**Technical Novelty And Significance:** 2
**Empirical Novelty And Significance:** 3
**Recommendation:** 6

**Clarity, Quality, Novelty And Reproducibility:**

The idea of this paper is presented clearly. The major novelty of this paper lies in the training strategy: pretraining using inaccurate and cheep data and then fine-tuning using accurate but expensive data. Considering DL model training in a broader scope, similar training schemes are seen in many domains. Thus, I rate the novelty of this paper's idea intermediate.

**Strength And Weaknesses:**

Strengths:
1. The training strategy proposed in this paper makes sense for ML MD. The authors also investigated data weighting based on quality and explored the use of unlabelled data for pretraining.
2. The experiments conducted in this research are thorough and their results support the main claim.

Weaknesses:
1. The authors should provide more details about the derivation of Eq. 3.
2. ML FF methods mentioned in the related work were not compared with the proposed method.


**Summary Of The Paper:**

The authors of this paper present a training strategy to improve performance in ML MD. In this strategy, first bias of inaccurate data is calculated using the model trained on accurate, then the bias-aware inaccurate data is used to train an ML MD model from scratch, and finally, the model is fine-tuned using accurate data. Pretraining using unlabeled data is also explored in this work.

**Summary Of The Review:**

Overall, this paper shows a carefully design pretraining-finetuning training strategy for ML MD, and their results support the claim that inaccurate data can help boost the performance of ML MD models as accurate data are very expensive to generate.

Update: I have read through all the reviewers' comments and the authors' response. I think the authors addressed most of the concerns. Even though I still have concerns about the novelty of this work, it may be an informative paper for researchers in the MLFF field. Thus, I increased my rating to 6: marginally above the acceptance threshold.

---

> ### Author Response · Authors · 2022-11-19
> **Thank you for the helpful feedback.**
>
> We appreciate your constructive suggestions. We provide our response to your comments as follows.
>
> **The authors should provide more details about the derivation of Eq. 3.:**
>
> Thank you for the suggestion! We have added a full derivation of Equation 3 to the appendix.
>
> **ML FF methods mentioned in the related work were not compared with the proposed method.:**
>
> **ANI-1ccx:** We have added these results to Table 4 (appendix A.7). In particular, we evaluate the provided ANI-1ccx checkpoint on the MD17 dataset both with and without fine tuning on the MD17 dataset. The results have been added to Table 4. Although ANI-1ccx was trained on over 20-million molecular conformations, our method is much more accurate while requiring significantly less and cheaper data.
>
> **Delta-ML:** This baseline is not suitable for our setting as it requires all data to have both inaccurate and force labels, which we do not have access to for much of the inaccurately labeled data. In addition, to generate CCSD(T) predictions with Delta-ML methods, we still need to calculate DFT forces, which will be much slower than ASTEROID.
>
> **The major novelty of this paper lies in the training strategy: pretraining using inaccurate and cheep data and then fine-tuning using accurate but expensive data. Considering DL model training in a broader scope, similar training schemes are seen in many domains. Thus, I rate the novelty of this paper's idea intermediate:**
>
> Our main contribution is identifying an important problem that has not been considered in previous literature and proposing a methodology to solve it. Previous MLFF works do not take the data generation budget into consideration. This is not practical, and we propose to reduce the cost by using less accurate data. We do not claim pre-training using inaccurate data as our main contribution, it is just a tool we use to reduce the data generation cost.
>
> Another contribution is that we propose to use score matching as a pre-training method. This has not been discussed in existing literature, and the topic may be of independent interest.

---

### Decision · Program_Chairs · 2023-01-20

**Decision:**

Reject

**Justification For Why Not Higher Score:**

The technical novelty is limited. Such a data augmentation method is quite commonly used in other fields (and there are many other even more fancy and effective methods to clean or debias the training data). The reviewers also raised their concerns on many other details (including the theoretical justification, experimental settings, complexity analysis, etc.).

**Justification For Why Not Lower Score:**

N/A

**Metareview: Summary, Strengths And Weaknesses:**

This paper aims to improve the training of machine learning force fields for MD simulations. ML force fields usually rely on accurate data, which is expensive to generate. The authors augment the training procedure with less accurate but cheaper data. This is achieved by first training a model with a limited amount of accurate data, which is then used to estimate the bias of the less accurate data. The data is reweighted according to this bias, followed by training a model on those samples.

Overall speaking, this paper investigates an interesting idea. However, according to the reviewers, the technical novelty is limited. Such a data augmentation method is quite commonly used in other fields (and there are many other even more fancy and effective methods to clean or debias the training data). The reviewers also raised their concerns on many other details (including the theoretical justification, experimental settings, complexity analysis, etc.) The authors did a good job in addressing some of them (and some reviewers updated their scores), however, the eventual consensus is that the paper is not that exciting yet, and is below the bar of ICLR.